# Evaluation of the diagnostic value of the Modified Evan's Blue Dye Test for assessing aspiration in tracheostomized critically ill patients: A systematic review and meta-analysis

Eszter Szőke [1,2,3], Ildikó Szántó[1,4], Caner Turan [1,2], Bence Szabó[1], Márton Papp[1,2], Gábor Horváth[1,5], László Zubek[1,2], Péter Hegyi [1,6,7], Zsolt Molnár[1,2,8], András Lovas [1,3]*

**1** Centre for Translational Medicine, Semmelweis University, Budapest, Hungary, **2** Department of Intensive Therapy, Semmelweis University, Budapest, Hungary, **3** Kiskunhalas Semmelweis Hospital the Teaching Hospital of the University of Szeged, Kiskunhalas, Hungary, **4** Hospitals of Szabolcs-Szatmár Bereg Country, Jósa András Hospital, Nyíregyháza, Hungary, **5** Department of Pulmonology, Semmelweis University, Budapest, Hungary, **6** Institute of Pancreatic Diseases, Semmelweis University, Budapest, Hungary, **7** Institute for Translational Medicine, Medical School, University of Pécs, Pécs, Hungary, **8** Department of Anesthesiology and Intensive Therapy, Poznan University, Poznan, Poland

* anlovas@gmail.com

## Abstract

Dysphagia and aspiration are common complications in tracheostomized critically ill patients. The Modified Evan's Blue Dye Test (MEBDT) is a non-invasive, bedside, adjunctive rule-in signal for aspiration when positive. However, its diagnostic accuracy compared to the gold standard Fiberoptic Endoscopic Evaluation of Swallowing (FEES) in these patients remains unclear. We conducted a systematic review and meta-analysis to evaluate the diagnostic accuracy of the MEBDT for the detection of aspiration in tracheostomized critically ill patients. Study protocol was prospectively registered on PROSPERO (CRD:42023479920). A systematic search was performed across three databases (PubMed, Cochrane Central, and Embase), followed by a systematic screening process against predetermined selection criteria and included studies that provided data on the sensitivity and specificity of MEBDT versus FEES. The risk of bias and the level of evidence certainty in the included papers were assessed by the QUADAS-2 tool and GRADE approach respectively. Six out of 2227 screened studies were included. We found that MEBDT had a high specificity 95.42% (95% CI [67.38%, 99.53%]) and positive predictive value 95% (95% CI [81, 100]). In conclusion, the MEBDT can serve as a bedside adjunctive tool that indicates a potential aspiration risk when the result is positive. However, if the result is negative, further diagnostic assessments, like FEES, are recommended for high-risk patients.

**Data availability statement:** All relevant data underlying the findings described in this study are fully available within the manuscript and its Supporting Information files.

**Funding:** This work was funded in the framework of the Ph.D. program of the Centre for Translational Medicine, Semmelweis University (Budapest, Hungary). No external funders or sponsors had any role in the design, data collection, analysis, interpretation, and manuscript preparation.

**Competing interests:** The authors have declared that no competing interests exist.

**Abbreviations:** CI, Confidence Interval; FEES, Fiberoptic Endoscopic Evaluation of Swallowing; FN, False Negative; FP, False Positive; GRADE, Grading of Recommendations Assessment, Development, and Evaluation; ICU, Intensive Care Unit; MEBDT, Modified Evan's Blue Dye Test; NPV, Negative Predictive Value; PEG, Percutaneous endoscopic gastrostomy; PPV, Positive Predictive Value; PRISMA, Preferred Reporting Items for Systematic Reviews and Meta-Analysis; ROB, Risk of Bias; TP, True Positive; TN, True Negative; VFSS, Videofluoroscopic Swallowing Study.

## Introduction

Dysphagia and aspiration are common and significant problems in tracheostomized critically ill patients. The risk of aspiration can be severely high (50–80%) [1] among these patients, which can lead to serious complications, such as aspiration pneumonia, which can increase the length of hospital and ICU stay, due to increased morbidity [2,3]. When the weaning process of the mechanical ventilation is finalized, decannulation and initiation of oral feeding are considered in tracheostomized patients [4]. Accordingly, screening for aspiration in patients with tracheostomy is crucial. Even more, as recurring aspiration following inadequate decision making can increase the rate of aspiration pneumonia and failed decannulation.

Various invasive, such as Fiberoptic Endoscopic Evaluation of Swallowing (FEES) [5], and non-invasive methods, such as the Videofluoroscopic Swallowing Study (VFSS) [6], and the Modified Evan's Blue Dye Test (MEBDT) [7] are described for assessing aspiration in tracheostomized patients. Considering the application of MEBDT in everyday practice is crucial. Access to FEES is limited due to scarce skilled human resources worldwide. Additionally, speech-language pathologists are not always competent to perform FEES independently in many countries. MEBDT is an easy-to-perform bedside approach that indicates early the signs of aspiration in patients with tracheostomy. It does not demand specific professional competencies, in contrast to other methods such as FEES where skills for endoscopy are required or VFSS where the investigation necessitates radiation exposure [8].

As patient transfer is required for VFSS, we decided to investigate the other two methods that can be applied at the bedside. According to the current literature, there is a missing link due to lack of data on the diagnostic accuracy and feasibility of MEBDT as compared to FEES in tracheostomized critically ill patients. The aim of this study was to evaluate the diagnostic accuracy of MEBDT in comparison with FEES in the assessment of aspiration in critically ill patients with tracheostomy.

## Methods

A systematic review and meta-analysis were conducted in accordance with the Preferred Reporting Items for Systematic Reviews and Meta-Analysis (PRISMA) guideline [9] (see Supplementary Table 2) to compare the diagnostic accuracy of MEBDT with FEES for aspiration screening among tracheostomized critically ill patients. The protocol of the study was registered on PROSPERO under CRD4202347992.

### Ethical approval

No ethical approval was required for this systematic review with meta-analysis, as all data were already published in peer-reviewed journals. No patients were involved in the design, conduct or interpretation of our study.

### Eligibility criteria

The Population, Index Test, Reference Test, Diagnosis (PIRD) framework was applied to contextualize the research question and conduct the systematic selection.

Eligibility criteria were defined by PIRD as follows; P: tracheostomized patients, I: MEBDT, R: FEES, D: aspiration. Randomised controlled trials, observational, prospective, and retrospective studies were included, whereas reviews, case reports, case series, and simple descriptive studies were excluded. Pediatric populations (patients under 18 years of age), patients without tracheostomy tubes, and patients with contraindications to MEBDT and FEES were excluded from the analysis.

### Information sources

The systematic search was conducted in three databases, PubMed, Cochrane Central, and Embase on 24th November, 2023, and repeated it with the same search key and strategy on 25th June, 2025.

### Search strategy

The following search key was framed for the systematic search, with no restriction, no filters, and no automatic translations: ("FEES" OR ("fiberoptic" AND endoscop* AND "evaluation") OR ("fiberoptic" AND endoscop* AND exam* AND swallow*) OR ("flexible" AND endoscop* AND evaluat*) OR ("Modified" AND "Evans" AND "blue" AND "dye" AND test*) OR "MEBDT" OR ("Evans" AND "blue" AND "dye" AND test*) OR ("blue" AND "dye" AND test*)) AND (tracheost* OR endotrach* OR ("artificial" AND "airway") OR cannula* OR intubat* OR ("percutaneous" AND "dilatational" AND "tracheostomy"))

Two independent investigators (E.S. and I.S.) independently evaluated the articles from 15th November to 24th November 2023. Disagreements were resolved by a third independent reviewer (C.T.).

### Selection process

The selection process was carried out by two independent reviewers (E.S and I.S), who first removed all duplicates, then screened articles by title and abstract, and finally selected relevant studies by full text. Publications were excluded based on predetermined selection criteria (see PROSPERO under CRD4202347992).

### Data collection process

Data of eligible articles [8,10,11–13,14] were collected independently by two investigators (E.S and I.S), using a standardized data collection sheet based on consensus by study authors. A third independent reviewer (C.T.) resolved any disagreements. The following data items were extracted: title, first author, year of publication, study design, patient characteristics and clinical state, index test characteristics, reference test characteristics, and diagnostic outcomes, specifically sensitivity and specificity.

### Risk of bias assessment

Two authors (E.S. and I.S.) performed the risk of bias assessment independently. On the basis of the recommendation of the Cochrane Collaboration, the QUADAS-2 risk of bias assessment tool was applied [15]. Two independent reviewers performed the assessment, and an independent third investigator resolved disagreements. Publication bias could not be assessed through visual inspection of Funnel-plots and Egger's test as less than 10 articles were selected. The level of certainty of evidence of the studies included was assessed with the GRADE-Pro tool [16], based on the recommendation of the Cochrane Collaboration.

### Synthesis methods

For the estimation of pooled specificity and sensitivity, the bivariate model of Chu et al (2006) [17] and Reitsma et al (2005) [18] was fitted. This approach considers the possible association between sensitivity and specificity. We plotted

individual and pooled sensitivities and specificities of studies included, their summary estimates, and the corresponding 95% confidence and prediction regions. In these visualizations, the sizes of ellipsoids reflect the weights of the studies calculated according to the method described by Burke et al (2018) [19]. For PPV and NPV, separate univariate analyses were performed using the generalized mixed-effect approach (Stijten et al, 2010) [20]. Heterogeneity was assessed by calculating I² measure and its confidence interval arising from the separate univariate analyses.

PPV and NPV values for different prevalences (ranging from 0% to 100%) were also calculated from the pooled specificity and sensitivity values, to examine how the effect of prevalence changes the predictive values. These calculations were performed to illustrate the dependence of predictive values on population prevalence. For these calculations the following formulae were utilized:

$$PPV = (Sens_{pooled} \times P)/((Sens_{pooled} \times P) + ((1 - Spec_{pooled}) \times (1 - P)))$$

$$NPV = (Spec_{pooled} \times (1 - P))/(Spec_{pooled} \times (1 - P) + (1 - Sens_{pooled}) \times P),$$

where $Sens_{pooled}$ and $Spec_{pooled}$ are the the pooled sensitivity and specificity values derived from the bivariate meta-analysis of diagnostic accuracy, and $P$ is the prevalence value.

Statistical analyses were carried out with R statistical software (version 4.1.2., R-core team, 2023) [21,22] using the meta (Balduzzi et al, 2019) [23] and the lme4 (Bates et al, 2015) [24] packages, and were partially based on the web-tool of Freeman et al (2019) [25]. The statistical analyses followed the advice of Harrer et al (2024 [26]).

## Results

### Search and selection

The systematic search identified 2,227 records from the databases; 1,819 unique records remained after duplicate removal (Fig 1). After title-abstract selection, 61 records remained, of which six studies [8,10,11–13,14] met the inclusion criteria. All studies were observational studies.

### Basic characteristics of studies included

Basic characteristics of the six articles included are presented in Table 1.

### Sensitivity and Specificity

On the basis of bivariate pooling of sensitivity and specificity, the overall sensitivity was 69.54% [95% CI: 37.68%, 89.60%] (Fig 2.). The overall pooled specificity derived from the six articles was 95.42% [67.38%, 99.53%] (Fig 2).

### Negative and positive predictive values

The prevalence of aspiration in the six articles ranged from 58.82% to 82.46% (Belafsky 73.33%, Filho 58.82%, Warnecke 60.98%, Fiorelli 82.46%, Munoz 70.73%, and Winklmaier 70%). Given this relatively wide prevalence range (23.64%), the pooled negative predictive value (NPV) was 59% [45%, 74%] (Fig 3).

The pooled overall positive predictive value (PPV) was 95% [81%, 100%] (Fig 4), based on the six selected articles.

Apart from the meta-analytic pooling of PPV and NPV values reported in the articles, these two metrics were also calculated from the above mentioned pooled specificity and sensitivity values. These calculated predictive values were then plotted against prevalence (Fig 5).

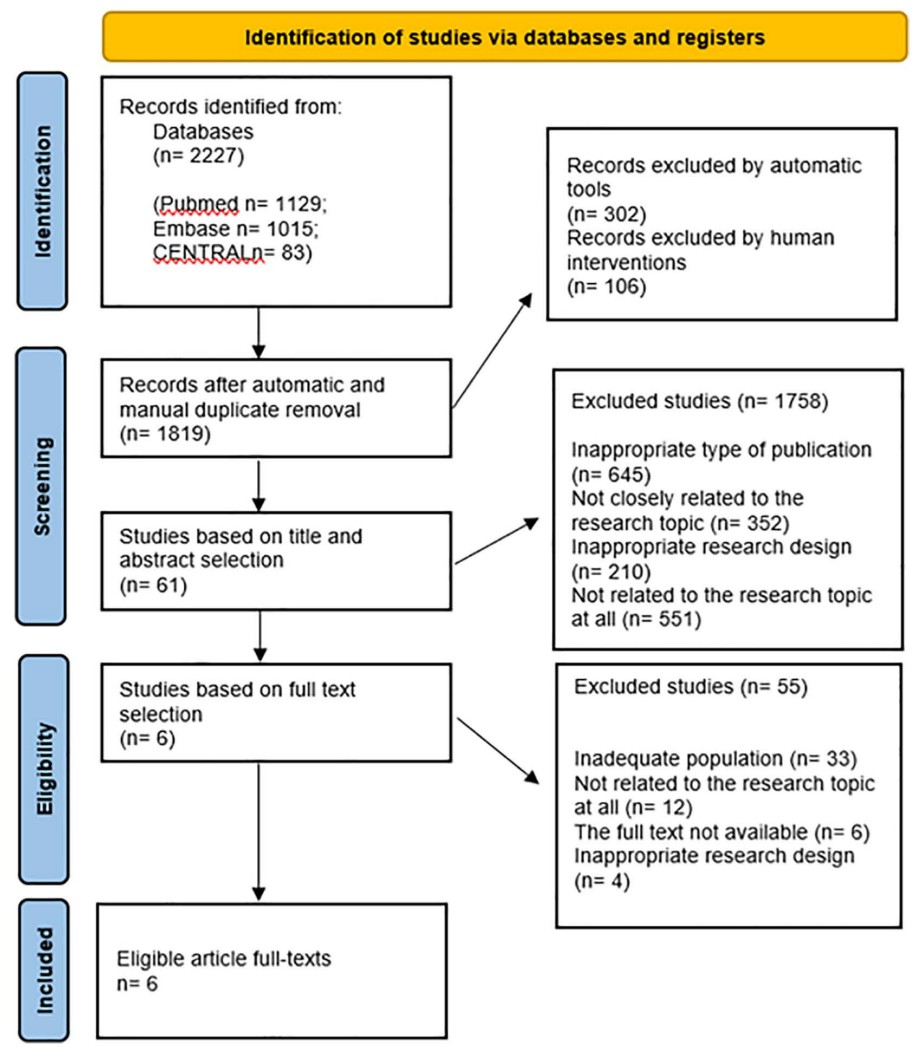

**Fig 1. The PRISMA flowchart of systematic selection.**

**Table 1. Basic characteristics of studies included.**

| First author | Year of publication | N0 of patients | Age (year) mean±SD | Study Type | Study period |
|---|---|---|---|---|---|
| Belafsky et al [8] | 2003 | 30 | 65±11 | Prospective | not specified |
| Filho et al [11] | 2019 | 17 | 60±21.0 | Observational | not specified |
| Winklmaieret al [12] | 2007 | 30 | 43-78 | Prospective | April 2005 – March 2006 |
| Warneckeet al [13] | 2013 | 41 | 56±15 | Prospective | within a one-year period (not specified) |
| Munoz-Garachet al [14] | 2023 | 41 | 61±14 | Prospective | not specified |
| Fiorelli et al [10] | 2016 | 57 | 67±6 | Retrospective | October 2013 – December 2015 |

## Publication bias and heterogeneity

To assess heterogeneity of the bivariate sensitivity and specificity pool, we performed a leave-one-out (LOO) analysis (S3 Fig.). On the basis of this "sensitivity" analysis, specificity was not affected by the omission of any article (LOO specificity

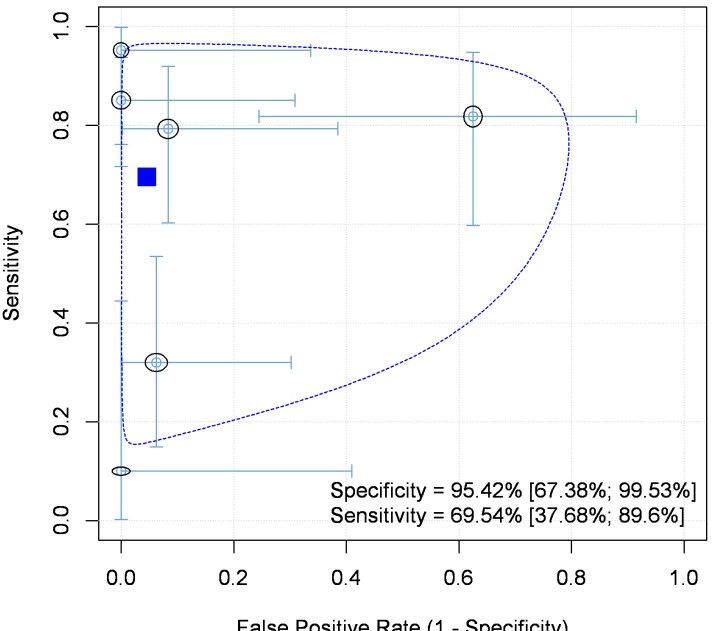

**Fig 2. Random-effects bivariate diagnostic meta-analysis of sensitivity and specificity of the Modified Evan's Blue Dye Test compared to the Fiberoscopic Evaluation of Swallowing in the diagnosis of aspiration.** Abbreviations: C*rosses around dots represent confidence intervals in both dimensions. The confidence region of 2D 95% is marked with blue dashed line.*

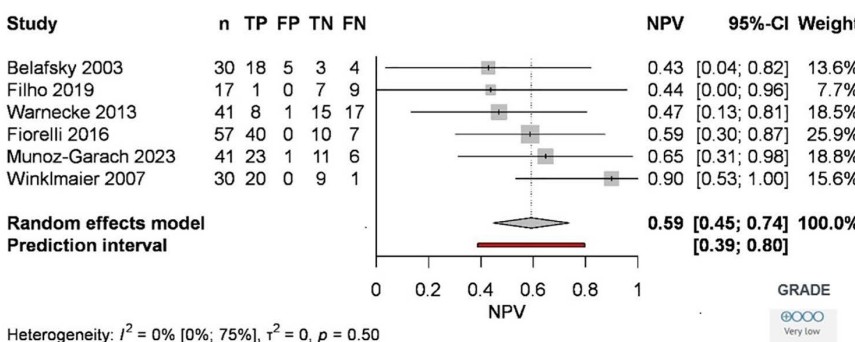

**Fig 3. Forest plot of the negative predictive value of the Modified Evan's Blue Dye Test compared to the Fiberoscopic Evaluation of Swallowing in the diagnosis of aspiration.** Abbreviations: *TP: true positive; FP: false positive; TN: true negative, FN: false negative; n: number of samples; CI: confidence interval.*

ranging from 93% to 95%), sensitivity was, however, somewhat affected by the omission of articles (LOO sensitivity ranges from 61% to 79%). Significant heterogeneity was identified among the studies in terms of univariate sensitivity ($I^2 = 94\%$, $p < 0.01$) and specificity ($I^2 = 72\%$, $p < 0.01$) (S1 and S2 Figs.).

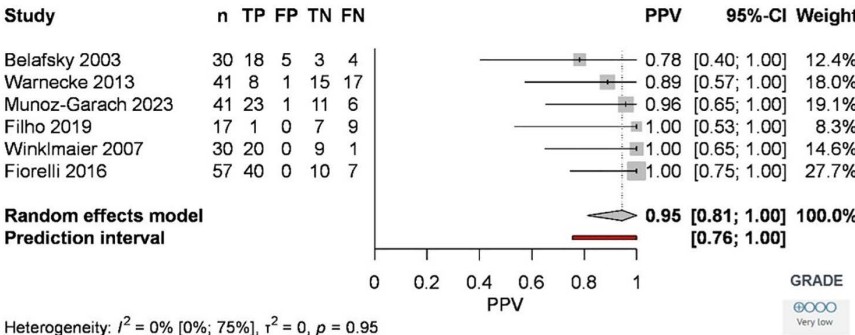

**Fig 4. Forest plot of the positive predictive value of the Modified Evan's Blue Dye Test compared to the Fiberoscopic Evaluation of Swallowing in the diagnosis of aspiration.** Abbreviations: *TP: true positive; FP: false positive; TN: true negative, FN: false negative; n: number of samples; CI: confidence interval.*

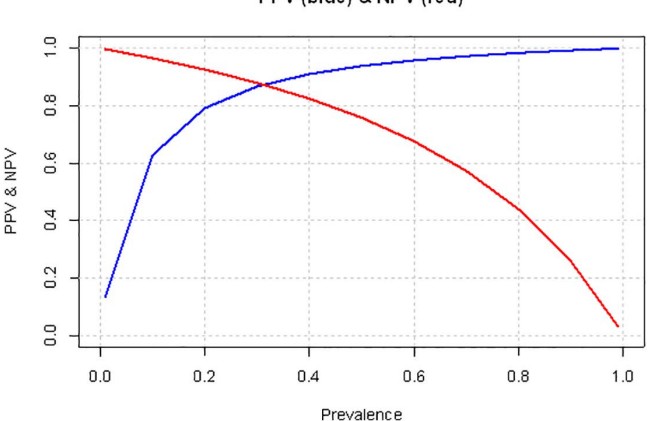

**Fig 5. Linegraph of the dynamics of positive and negative predictive in relation to changing prevalences.**

## Risk of bias and GRADE assessment

The methodological quality was assessed by determining the risk of bias across four domains of the QUADAS-2 evaluation tool [15] for diagnostic studies. During the analysis, the risk of bias was assessed as low in two of the articles, and high in four. In terms of GRADE assessment [27], the overall risk of bias was considered as serious due to the inclusion of a paper by Belafsky et al. [8] pooled for outcome, and the high risk of bias observed in the flow and timing domains of four articles [8,10,12,14] (Table 2). We found that certainty of evidence was low, the most important issue was diagnostic accuracy and the effect of the test due to inconsistency and variability between different methods. GRADE results are shown in forest plots (S1 and S2 Figs.; Figs 3 and 4).

## Discussion

In intensive care units, swallowing assessment of patients with tracheostomy tubes can promote individualized decision-making by excluding or confirming aspiration, as well as determining the most appropriate nutritional consistency for the patient. MEBDT may serve as an adjunctive bedside test with high specificity and positive predictive value. When the result is positive, it indicates a strong likelihood of aspiration; however, a negative result cannot reliably rule out the possibility of aspiration.

**Table 2. Summary of risk of bias and applicability concerns assessed by review authors for each domain in the included studies.**

| Studies | Patient selection | Index test | Reference standard | Flow & timing | Overall |
|---------|-------------------|------------|--------------------|--------------|---------|
| Belafsky et al., 2003 [8] | low | high | high | high | high |
| Munoz-Garach et al., 2023 [14] | low | low | low | high | high |
| Warnecke et al., 2013 [13] | low | low | low | low | low |
| Filho et al., 2019 [11] | low | low | low | low | low |
| Winklmaier et al., 2007 [12] | low | low | low | high | high |
| Fiorelli et al., 2016 [10] | low | low | low | high | high |

### Comparison of FEES and MEBDT: Advantages and practical considerations

MEBDT has already been performed with food materials, such as ice, liquid, or mashed food, and blue dye. During the application of the MEBDT, the cuff of the tracheostomy tube is deflated. The patient takes blue-colored food or a drink of varying textures and amounts orally and is then asked to swallow. The MEBDT is positive if colored secretion appears when the trachea is suctioned through the tracheal cannula [10].

In contrast, FEES is an invasive procedure in which a thin fiberoptic endoscope is inserted transnasally to visualize the epiglottis, vocal cords, and the pharyngeal phase of swallowing in real-time. The procedure enables the detection of residue in the piriform sinuses and facilitates the assessment of movement disorders affecting the vocal cords and epiglottis. Local anesthesia is generally avoided, as it may compromise the sensory aspects of swallowing and increase the risk of aspiration. [28–30]. FEES is positive for aspiration if saliva, food, or any liquid is trapped under the vocal cords [31]. The blue coloration of the swallowed material can further enhance the assessment of aspiration. Both FEES and MEBDT can be performed at the bedside. One of the main advantages of FEES is that it is particularly useful in differentiating the pathophysiological background of dysphagia-related aspiration [32]. However, it may not be readily available in all medical settings, as it requires special instrument and skilled personnel trained in its use [33]. In contrast, MEBDT is much simpler and does not require specialized endoscopic skills or advanced professional training, [7,34] or special instruments. This can be a significant advantage in settings where endoscopy is not readily available.

### Methodology and implication variability in MEBDT

The design of MEBDT in the studies included differed significantly (S1 Table). All six articles used different consistencies and amounts of food and liquid, and the time intervals between the index and the reference tests were also variable. Time frames for tracheal aspiration also varied widely. Without standardization, it is difficult to establish a proper protocol for MEBDT.

An important factor to consider during MEBDT is the temperature of the food or fluid being administered. Belafsky et al [8] used cold temperature bolus using ice chips. The sensation of cold may enhance swallowing reflexes toward improved motor control, and at the same time, induce rapid contraction of the esophageal muscles [35–38]. This condition may influence the swallowing test, and therefore room temperature fluid and food administration should be preferred in future studies.

MEBDT was performed with 100% PPV in three studies [10-12]. This suggests the application of different consistencies (water, saliva, or honey-like), different (5-10-20 ml) and repeated quantities of material to be swallowed. In this case, a PPV of 100% indicates that, in these studies, all patients who tested positive for aspiration using MEBDT were confirmed to have aspiration, meaning that all positive MEBDT findings represented true aspiration events across the tested bolus consistencies, reflecting high reliability for aspiration confirmation rather than overall diagnostic accuracy.

The time lag between MEBDT as an index and FEES as a reference test should be considered, as it might affect the reliable comparison of the two tests. Half of the articles did not perform the tests within one session [8,10,12,14] (S1 Table). In our opinion, a considerable delay between MEBDT and FEES could influence the diagnostic accuracy. This is because significant changes can be observed in the state of critically ill patients, even within a few hours. Therefore, the application of both tests in one session is recommended for future studies.

Only one study [8] included mechanically ventilated patients with minimal pressure support (10 out of 30 patients). Mechanical ventilation, even with minimized pressures could significantly affect diagnostic accuracy, as in the study by Belafsky et al [8], where the sensitivity of MEBDT was 100% in the case of mechanically ventilated patients. As the cuff was deflated, air leakage should be assumed with positive pressure support ventilation. Airflow from the trachea, to and through the vocal cords could influence aspiration. Therefore, we recommend the inclusion of patients weaned from mechanical ventilation in future studies, but further studies are also needed on the evaluation of aspiration of MEBDT with mechanical ventilation.

Among the articles, Filho et al [11] and Warnecke et al [13] reported the highest number of false negative cases relative to the total number, 9 and 17, respectively (Fig 4). However, the reasons are not discussed in the articles, but a few factors may be hypothesized to be behind these findings. One can be the limitations of the examination technique. When such a small amount of food or liquid is aspirated that is only visible by the fiberoptic examination, we speak of microaspiration [39]. In this case, the amount aspirated is so small that it cannot be suctioned from the trachea, even after repeated attempts.

Furthermore, there is no literature on the optimal amount of blue dye to add to food or liquid. Low concentrations may be too diluted by tracheal secretions, making aspiration less visible during tracheal suction.

Therefore, to improve the diagnostic accuracy of MEBDT, we recommend the use of high-concentration blue color and repeated attempts of tracheal suction. Blue dye should always be used, as other colors like red or green might be mistaken for blood or pus. Using a transparent suction tube with a white paper background helps make blue-colored tracheal secretions more visible during suction.

## Effectiveness and limitations of MEBDT in aspiration detection: Findings from meta-analysis

Our meta-analysis has revealed that MEBDT, despite its lower sensitivity (69.5%), has a high positive predictive value (95%) that makes it a useful, adjunctive, bedside investigation for detecting aspiration in tracheostomized patients. This high PPV suggests that MEBDT is effective in confirming aspiration when the result is positive, particularly in a population with a higher prevalence of aspiration. However, although MEBDT has a high specificity (95%), its negative predictive value (NPV) is relatively low at 59%. This means that although a positive result is reliable, a negative result is less so for ruling out aspiration. Therefore, FEES is recommended as the next step following a negative MEBDT result for accurate detection.

It is important to emphasize that the calculated positive and negative predictive values (PPV and NPV) across different prevalence rates are presented to facilitate the interpretation of predictive performance. The PPV was consistently high throughout most of the plausible prevalence range. Even at a prevalence of 20%, the PPV was already around 80%, and for prevalences above 40%, the PPV consistently exceeded 90% (see Fig. 5). It should be emphasized that PPV and NPV, and the conclusions based on them, are only valid for populations with prevalence rates similar to those observed in the included studies (approximately 60–80%, which already represent a relatively heterogeneous range). Therefore, conclusions based on these predictive values cannot be directly applied to populations with substantially lower or higher prevalence.

## Strengths and limitations

This is the first systematic review and meta-analysis comparing MEBDT with FEES, supplemented with RoB and GRADE assessments. This study directly informs bedside practice on the possibilities of using MEBDT in tracheostomized

patients, and due to its simplicity, it can be immediately transferred into clinical practice. The primary limitation of this study is the relatively small total number of registered cases (n = 216). Additionally, only six articles were included, all of which were observational studies, resulting in substantial heterogeneity in clinical practice. A clearly defined study period was reported in only two of the six included studies, limiting methodological transparency and comparability (S1 Table). In addition, while FEES represents a standardized reference method, MEBDT lacks a uniformly standardized protocol, which may have contributed to methodological variability.

### Implications for practice and research

Currently, due to methodological heterogeneity and limited evidence, MEBDT can serve as a bedside rule-in signal for aspiration in critically ill tracheostomized patients when positive; however, negative results require confirmation with an instrumental assessment. Additionally, there is insufficient data on the reliability of MEBDT in mechanically ventilated patients. The results from our systematic review and meta-analysis help inform clinical decision-making for dysphagia assessment in tracheostomized patients, highlighting areas where further research is needed. Future research should focus on prospective, standardized studies to further specify the sensitivity, specificity, and positive predictive value of MEBDT, taking into account the influence of aspiration prevalence and the possible heterogeneity of the investigating method. To establish MEBDT as a reliable standard screening test for aspiration, future research should aim to implement a standardized, reproducible protocol, and validate its performance prospectively across diverse patient populations against the gold standard FEES.

## Conclusion

The Modified Evan's Blue Dye Test (MEBDT) may serve as an adjunctive bedside tool to identify aspiration in tracheostomized patients. A positive MEBDT result, supported by high specificity and positive predictive value – which may be influenced by the high prevalence of aspiration in the included studies – can indicate the presence of aspiration. However, the moderate pooled sensitivity and low negative predictive value indicate that a negative MEBDT result cannot safely exclude aspiration, and therefore MEBDT should not be considered as a screening test. Negative results should prompt further instrumental assessment, such as FEES, particularly in high-risk patients.

### Key points

1. MEBDT can serve as a rule-in bedside signal for aspiration when positive.

2. MEBDT has high positive predictive value for confirming aspiration when positive, but negative results cannot reliably exclude aspiration.

3. Negative MEBDT result cannot reliably rule out aspiration requiring further diagnostic assessments.

## Supporting information

**S1 Fig. Sensitivity of the Modified Evan's Blue Dye Test compared to the Fiberoscopic Evaluation of Swallowing in the diagnosis of aspiration.**
(TIF)

**S2 Fig. Specificity of the Modified Evan's Blue Dye Test compared to the Fiberoscopic Evaluation of Swallowing in the diagnosis of aspiration.**
(TIF)

**S3 Fig. Leave-one-out (LOO) analysis.**
(TIF)

**S4 Fig. Random-effects bivariate diagnostic meta-analysis of NPV and PPV of the Modified Evan's Blue Dye Test compared to the Fiberoscopic Evaluation of Swallowing in the diagnosis of aspiration.**
(TIF)

**S1 Table. Methodological assessment of studies included.**
(DOCX)

**S2 Table. PRISMA 2020 Checklist.**
(DOCX)

**S3 Table. Data extraction_MEBDT.**
(XLSX)

## Author contributions

**Conceptualization:** Eszter Szőke, Caner Turan, Gábor Horváth, László Zubek, Péter Hegyi, Zsolt Molnár, András Lovas.

**Data curation:** Eszter Szőke, Ildikó Szántó, András Lovas.

**Formal analysis:** Eszter Szőke, Ildikó Szántó, Bence Szabó.

**Funding acquisition:** Péter Hegyi.

**Investigation:** Eszter Szőke, Ildikó Szántó, András Lovas.

**Methodology:** Eszter Szőke, Caner Turan, Bence Szabó, Péter Hegyi, Zsolt Molnár, András Lovas.

**Project administration:** Péter Hegyi, András Lovas.

**Resources:** Péter Hegyi.

**Supervision:** Caner Turan, Gábor Horváth, László Zubek, Péter Hegyi, Zsolt Molnár, András Lovas.

**Validation:** Zsolt Molnár.

**Visualization:** Bence Szabó.

**Writing – original draft:** Eszter Szőke, Caner Turan, András Lovas.

**Writing – review & editing:** Eszter Szőke, Ildikó Szántó, Caner Turan, Bence Szabó, Márton Papp, Gábor Horváth, László Zubek, Péter Hegyi, Zsolt Molnár, András Lovas.

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
