## [Decision Letter · Decision Letter 0]

9 Dec 2025

PONE-D-25-53361The Modified Evan’s Blue Dye Test Has High Positive Predictive Value for Diagnosing Aspiration in Tracheostomized Critically Ill Patients: A Systematic Review and Meta-Analysis.PLOS One

Dear Dr. Lovas,

Thank you for submitting your manuscript to PLOS ONE. After careful consideration, we feel that it has merit but does not fully meet PLOS ONE’s publication criteria as it currently stands. Therefore, we invite you to submit a revised version of the manuscript that addresses the points raised during the review process.

We look forward to receiving your revised manuscript.

Kind regards,

Jeyasakthy Saniasiaya, MD, MMed ORLHNS, FEBORLHNS

Academic Editor

PLOS One

Journal Requirements:

2. Peer review at PLOS  One is not double-blinded (https://journals.plos.org/plosone/s/editorial-and-peer-review-process). For this reason, authors should include in the revised manuscript all the information removed for blind review.

3. Please include your tables as part of your main manuscript and remove the individual files. Please note that supplementary tables (should remain/ be uploaded) as separate "supporting information" files.

“This work was funded in the framework of the Ph.D. program of the Centre for Translational Medicine, Semmelweis University (Budapest, Hungary). No external funders or sponsors had any role in the design, data collection, analysis, interpretation, and manuscript preparation.”

“This work was funded in the framework of the Ph.D. program of the Centre for Translational Medicine, Semmelweis University (Budapest, Hungary). No external funders or sponsors had any role in the design, data collection, analysis, interpretation, and manuscript preparation.”

6. PLOS requires an ORCID iD for the corresponding author in Editorial Manager on papers submitted after December 6th, 2016. Please ensure that you have an ORCID iD and that it is validated in Editorial Manager. To do this, go to ‘Update my Information’ (in the upper left-hand corner of the main menu), and click on the Fetch/Validate link next to the ORCID field. This will take you to the ORCID site and allow you to create a new iD or authenticate a pre-existing iD in Editorial Manager.

7. Please remove your figures from within your manuscript file, leaving only the individual TIFF/EPS image files, uploaded separately. These will be automatically included in the reviewers’ PDF.

Additional Editor Comments:

Please highlight how this study adds to the current literature

Reviewers' comments:

Reviewer's Responses to Questions

**Comments to the Author**

1. Is the manuscript technically sound, and do the data support the conclusions?

Reviewer #1: Partly

Reviewer #2: Yes

2. Has the statistical analysis been performed appropriately and rigorously? 

Reviewer #1: Yes

Reviewer #2: Yes

3. Have the authors made all data underlying the findings in their manuscript fully available?

Reviewer #1: Yes

Reviewer #2: Yes

4. Is the manuscript presented in an intelligible fashion and written in standard English?

Reviewer #1: No

Reviewer #2: Yes

5. Review Comments to the Author

Reviewer #1: Thank you for your time in presenting this manuscript. My thoughts are as follows.

In the abstract, the authors claim that "[they] conducted a systematic review and meta-analysis to investigate the effectiveness of MEBDT in detecting aspiration in tracheostomized critically ill patients." However, they say that the, "aim of this study was to evaluate the diagnostic accuracy of MEBDT in comparison with FEES in the assessment of aspiration in critically ill patients with tracheostomy." While similar, these two aims are different. They should either be addended to be the same, or a primary and secondary aim should have been set a priori to the study.

There are other areas of the manuscript where the authors seem to contradict themselves, make incorrect statements, or clarifying statements are warranted for a number of reasons.

1. page 13, line 76: "MEBDT should be performed in all critically ill tracheostomized patients." However, authors state that, "It is important to emphasize that PPV and NPV, and the conclusions based on them, are only valid for populations where the prevalence is similar to those in the studies examined (between 60-80%, which already represent quite heterogeneous prevalences). Therefore, conclusions based on MS do not apply to patient populations where the prevalence of aspiration is, for example, 10% (where PPV would be around 50% and NPV around 90%) or to a population with 90% prevalence (PPV ~100%, NPV ~20%)," on page 23. This is a major caveat, and a major flaw of their listed key point.

2. page 26, lines 303-308: "This suggests the application of different consistencies (water, saliva, or honey-like), different (5-10-20 ml) and repeated quantities of material to be swallowed. In this case, a PPV of 100% indicates that, in these studies, all patients who tested positive for aspiration using MEBDT were confirmed to have aspiration, meaning that the test identifies aspirates of such consistencies with high accuracy." This is not what it means. Accuracy can not be explained by one parameter like PPV. For instance, this test has a high PPV but a moderate sensitivity, indicating that it's not perfect at detecting all positive cases. This is an important fact and it is not consistent with a "high accuracy" at all.

3. page 23, line 242: "However, for NPV and PPV, heterogeneity values were 0% with p = 0.5 and p = 0.95, respectively (Figs. 3 and 4). However, due to the low number of articles, these metrics do not provide much relevant information." I don't know what to say about this because at least half of the author's argument throughout the paper is that the positive predictive value being near 100% demonstrates high accuracy and/or is the basis for their argument in favor of widespread MEBDT use as a screening tool. What is the purpose of writing a paragraph about a heterogeneity analysis that "does not provide much relevant information."

4. The authors reference the use of local anesthetic during FEES. While it is true that local anesthetic may be used during the procedure, it is not necessary and should be counted against its use in clinical practice. Some literature and practice experts suggests that anesthesia worsens the diagnostic ability of FEES, and recommend against its use.

There are numerous places in the manuscript that are clearly missing information/words or used inconsistent formatting. Some examples included but are not limited to

1. page 12, lines 61-62: "Six out of XXX screened studies were included," where the XXX was not updated.

2. page 16, lines 124-125: "Pediatric population, patients without tracheostomy tube, and patients with contraindications to MEBDT and FEES were excluded," where "population" and "tube" should be plural.

3. page 16, lines 129-130: "...on 24th November 2023, and repeated it with the same search key and strategy on June 25, 2025."

4. page 25, lines 279-280: "FEES is positive for aspiration if saliva, food, or any liquid trapped under the vocal cords [28]," where the word "is" is missing between liquid and trapped.

There are many other places in the manuscript that were understandable, but clearly not written to the standard of an accredited scientific journal publication. I realize the authors are not from a predominately English-speaking country, but these issues should be addressed.

1. page 14, lines 103-104: "As transfer of the patient is inevitable for VFSS, our decision was to investigate the other two methods, applicable at the bedside."

2. page 28, lines 353-355: As for limitations, only a small number of cases were registered, 216 in total. In addition, only six articles could be included, all of which were observational studies, and reported significantly heterogenous practices."

While the peer review process is meant to help improve the quality of submitted manuscripts, these points made it very difficult to want to keep reading further. These issues should make anyone second-guess the quality of the data and conclusions being presented. Therefore, I would not recommend this article for publication in its current state. If this is to be considered, the article would require intensive edits on sentence structure. It may be useful to enlist the support of a native English writing service for this manuscript, as there is a lot of potential in the work.

Reviewer #2: This is a systematic review and meta-analysis to investigate the effectiveness of MEBDT in detecting aspirations in tracheostomized critically ill patients. Search was performed across three databases (PubMed, Cochrane Central, and Embase), on November 25, 2023, and repeated with the same search key and strategy on June 25, 2025. This was followed by a systematic screening process against predetermined selection. The aim of this study was to evaluate the diagnostic accuracy of MEBDT in comparison with FEES in the assessment of aspiration in critically ill patients with tracheostomy. It was hypothesized and aimed to test whether MEBDT is at least as accurate as FEES in diagnosing aspiration. The main finding was that MEBDT had a high specificity 95.42% (95% CI [67.38%, 99.53%]) and positive predictive value 95% (95% CI [81, 100]).

After a review of the manuscript, I have the following observations,

Methods: Looking at the 6 studies included in the review, there was only 2/6 of the studies that had a study period, which were Winklmaier et al and Fiorelli et al, others were either unstated or non-specific.

A pooled negative predictive value of 59% is indicative of a variable sensitivity of the use MEBDT which is a major limitation. When a negative result, i.e no blue dye is found, rather than completely ruling out aspiration, it may be due to presence of small aspirations or silent aspirations, where there are no cough reflex present. As well as the high specificity reported, 95.42% (95% CI [67.38%, 99.53%]) merely shows the ability to detect patients who do not aspirate, rather than being as good a diagnostic tool as FEES. This is because FEES is standardized in its uses as opposed to MEBDT, which makes comparism and drawing conclusions about studies with a variety of protocols quite challenging.

The risk of bias and GRADE assessment was high for 4/6 papers, which raises issues and concerns about the applicability of the findings of these publications.

6. PLOS authors have the option to publish the peer review history of their article (what does this mean?). If published, this will include your full peer review and any attached files.

Reviewer #1: No

Reviewer #2: **Yes:** Stephen O Adebola

---

## [Author Response · Author response to Decision Letter 1]

28 Jan 2026

Response to Reviewers

We appreciate the reviewers for their careful reading of our manuscript and for their valuable and constructive suggestions, which have helped us improve the clarity and accuracy of our work. All major revisions have been highlighted in grey in the revised manuscript (“Revised Manuscript With Changes Highlighted”). Deleted text has been retained using strikethrough where appropriate.

Reviewer 1

Comment

Reviewer’s comment:

“In the abstract, the authors claim that "[they] conducted a systematic review and meta-analysis to investigate the effectiveness of MEBDT in detecting aspiration in tracheostomized critically ill patients." However, they say that the, "aim of this study was to evaluate the diagnostic accuracy of MEBDT in comparison with FEES in the assessment of aspiration in critically ill patients with tracheostomy." While similar, these two aims are different. They should either be addended to be the same, or a primary and secondary aim should have been set a priori to the study.”

Response:

We thank the reviewer for this important and constructive comment. We agree that the original wording suggested two slightly different study aims. To address this issue, we revised the manuscript to consistently define a single and clear a priority objective. Specifically, we aligned the abstract with the stated aim of the study and clarified that the primary objective of this systematic review and meta-analysis was to evaluate the diagnostic accuracy of the Modified Evan’s Blue Dye Test (MEBDT) in comparison with Fiberoptic Endoscopic Evaluation of Swallowing (FEES) for the assessment of aspiration in tracheostomized critically ill patients. The term “effectiveness” was replaced to ensure conceptual clarity and to accurately reflect the diagnostic test accuracy framework of the study.

Changes made:

• The aim-related sentence in the abstract was revised by replacing “effectiveness” with terminology reflecting diagnostic accuracy compared with FEES.

• The aim of the study is now stated consistently across the abstract and the main manuscript (page 4, lines 53-57).

Comment 1

Reviewer’s comment:

“1. page 13, line 76: "MEBDT should be performed in all critically ill tracheostomized patients." However, authors state that, "It is important to emphasize that PPV and NPV, and the conclusions based on them, are only valid for populations where the prevalence is similar to those in the studies examined (between 60-80%, which already represent quite heterogeneous prevalences). Therefore, conclusions based on MS do not apply to patient populations where the prevalence of aspiration is, for example, 10% (where PPV would be around 50% and NPV around 90%) or to a population with 90% prevalence (PPV ~100%, NPV ~20%)," on page 23. This is a major caveat, and a major flaw of their listed key point.”

Response:

We thank the reviewer for this important and well-founded comment. We agree that the original wording of the key point may have been overly broad and did not sufficiently reflect the important limitations related to disease prevalence and the interpretation of predictive values. To address this concern, we revised the relevant section to clarify the prevalence-dependent nature of PPV and NPV and to avoid overly generalized recommendations regarding the use of MEBDT. The revised text now explicitly emphasizes that predictive values should be interpreted within the context of prevalence ranges comparable to those observed in the included studies and that conclusions cannot be directly extrapolated to populations with substantially lower or higher aspiration prevalence. This modification ensures consistency between the key points and the methodological limitations discussed later in the manuscript.

Changes made:

• The statement suggesting universal application of MEBDT was revised to avoid overgeneralization and to reflect the prevalence-dependent interpretation of predictive values.

• The section discussing PPV and NPV was expanded and clarified to explicitly highlight the dependence of predictive values on disease prevalence and the limited generalizability of these results to populations with substantially different prevalence rates (page16, lines 255-269 & page page 22, lines 389-398).

• The key message was revised to ensure full consistency with the methodological considerations discussed in the Discussion section.

Comment 2

Reviewer’s comment:

„2. page 26, lines 303-308: "This suggests the application of different consistencies (water, saliva, or honey-like), different (5-10-20 ml) and repeated quantities of material to be swallowed. In this case, a PPV of 100% indicates that, in these studies, all patients who tested positive for aspiration using MEBDT were confirmed to have aspiration, meaning that the test identifies aspirates of such consistencies with high accuracy." This is not what it means. Accuracy can not be explained by one parameter like PPV. For instance, this test has a high PPV but a moderate sensitivity, indicating that it's not perfect at detecting all positive cases. This is an important fact and it is not consistent with a "high accuracy" at all.”

Response:

We thank the reviewer for this important and technically well-founded comment. We fully agree that overall diagnostic accuracy cannot be inferred from a single parameter such as positive predictive value alone, and that high PPV does not imply high sensitivity or comprehensive case detection. To address this issue, we revised the manuscript to avoid the term “high accuracy” when referring solely to PPV. The revised text now clearly distinguishes between overall diagnostic accuracy and the reliability of positive test results. Specifically, we clarify that a PPV of 100% indicates that all positive MEBDT findings in these studies represented true aspiration events across the tested bolus consistencies, reflecting high reliability for aspiration confirmation rather than overall diagnostic accuracy.

Changes made:

• The wording implying “high accuracy” based solely on PPV was removed.

• The text was revised to explicitly state that PPV reflects the reliability of positive test results and does not capture overall diagnostic accuracy or sensitivity (page 20-21, lines 343-346).

• The interpretation of MEBDT performance was revised to align with standard diagnostic test accuracy principles, particularly by avoiding the interpretation of PPV as a measure of overall diagnostic accuracy.________________________________________

Comment 3

Reviewer’s comment:

„3. page 23, line 242: "However, for NPV and PPV, heterogeneity values were 0% with p = 0.5 and p = 0.95, respectively (Figs. 3 and 4). However, due to the low number of articles, these metrics do not provide much relevant information." I don't know what to say about this because at least half of the author's argument throughout the paper is that the positive predictive value being near 100% demonstrates high accuracy and/or is the basis for their argument in favor of widespread MEBDT use as a screening tool. What is the purpose of writing a paragraph about a heterogeneity analysis that "does not provide much relevant information."

Response:

We thank the reviewer for this thoughtful and well-reasoned comment. We agree that presenting a heterogeneity analysis while simultaneously stating that it does not provide relevant information may be confusing and does not add meaningful interpretative value, especially in the context of a limited number of included studies. To address this concern, we removed this paragraph from the manuscript. This revision avoids potential inconsistency and improves the clarity and coherence of the results section by focusing on analyses that meaningfully contribute to the interpretation of MEBDT performance.

Changes made:

The paragraph describing heterogeneity analyses of PPV and NPV with limited interpretative value was removed from the Results section (page 17, lines 278-280).

Comment 4

Reviewer’s comment:

„4. The authors reference the use of local anesthetic during FEES. While it is true that local anesthetic may be used during the procedure, it is not necessary and should be counted against its use in clinical practice. Some literature and practice experts suggests that anesthesia worsens the diagnostic ability of FEES, and recommend against its use.”

Response:

We thank the reviewer for the comment regarding the use of local anesthesia during FEES. In accordance with the literature and clinical guidelines, we have revised the text to clarify that anesthesia is generally avoided during the procedure, as it may compromise the sensory aspects of swallowing and increase the risk of aspiration.

Changes made:

The sentence describing the use of local anesthesia during FEES was revised (page 19, lines 308-315) to clarify that anesthesia is generally avoided, as it may impair sensory assessment of swallowing and increase the risk of aspiration.

Comment

Reviewer’s comment:

„There are numerous places in the manuscript that are clearly missing information/words or used inconsistent formatting. Some examples included but are not limited to

1.page 12, lines 61-62: "Six out of XXX screened studies were included," where the XXX was not updated.

2.page 16, lines 124-125: "Pediatric population, patients without tracheostomy tube, and patients with contraindications to MEBDT and FEES were excluded," where "population" and "tube" should be plural.

3.page 16, lines 129-130: "...on 24th November 2023, and repeated it with the same search key and strategy on June 25, 2025."

4.page 25, lines 279-280: "FEES is positive for aspiration if saliva, food, or any liquid trapped under the vocal cords [28]," where the word "is" is missing between liquid and trapped.”

Response:

We thank the reviewer for carefully identifying these issues. We have thoroughly revised the manuscript to correct missing information, grammatical inaccuracies, and formatting inconsistencies. All examples raised by the reviewer were addressed, and the relevant sections were edited accordingly to improve clarity and accuracy.

Changes made:

•The total number of screened studies was updated to 2227 (“Six out of 2227 screened studies were included”) (page 4).

•Pluralization and wording were corrected in the exclusion criteria to improve grammatical accuracy and clarity (page 8).

•The description of the search strategy was revised to clearly state the initial search date and the subsequent repeated search using consistent terminology and formatting (page 8).

•The missing verb in the definition of aspiration during FEES was added (“…any liquid is trapped under the vocal cords”) (page 19).________________________________________

Comment

Reviewer’s comment:

“There are many other places in the manuscript that were understandable, but clearly not written to the standard of an accredited scientific journal publication. I realize the authors are not from a predominately English-speaking country, but these issues should be addressed.

1. page 14, lines 103-104: "As transfer of the patient is inevitable for VFSS, our decision was to investigate the other two methods, applicable at the bedside."

2. page 28, lines 353-355: As for limitations, only a small number of cases were registered, 216 in total. In addition, only six articles could be included, all of which were observational studies, and reported significantly heterogenous practices."”

Response:

We thank the reviewer for this constructive comment. We agree that improvements in language clarity and academic style were necessary. As we did not have access to a native English-speaking editor, the manuscript was carefully revised by the authors to enhance grammatical accuracy, sentence structure, and overall readability, with particular attention to the examples highlighted by the reviewer. The cited sentences were rewritten to meet the linguistic and stylistic standards expected of an accredited scientific journal.

Changes made:

• The sentence referring to patient transfer for VFSS was rewritten for improved clarity and academic tone (page 6, lines 106-108).

• The limitations section was revised and restructured to improve clarity, grammatical correctness, and precision while preserving the original content (page 23, lines 403-411).________________________________________

Reviewer 2

Comment

Reviewer’s comment:

“Methods: Looking at the 6 studies included in the review, there was only 2/6 of the studies that had a study period, which were Winklmaier et al and Fiorelli et al, others were either unstated or non-specific. A pooled negative predictive value of 59% is indicative of a variable sensitivity of the use MEBDT which is a major limitation. When a negative result, i.e no blue dye is found, rather than completely ruling out aspiration, it may be due to presence of small aspirations or silent aspirations, where there are no cough reflex present. As well as the high specificity reported, 95.42% (95% CI [67.38%, 99.53%]) merely shows the ability to detect patients who do not aspirate, rather than being as good a diagnostic tool as FEES. This is because FEES is standardized in its uses as opposed to MEBDT, which makes comparism and drawing conclusions about studies with a variety of protocols quite challenging.

The risk of bias and GRADE assessment was high for 4/6 papers, which raises issues and concerns about the applicability of the findings of these publications.”

Response:

We thank the reviewer for this comprehensive and insightful comment. We agree with these concerns and have revised the manuscript to more clearly and explicitly acknowledge the methodological limitations and diagnostic constraints highlighted. Specifically, the Limitations section was expanded to address the small overall sample size, the observational nature of the included studies, the substantial heterogeneity in clinical practice, and the fact that a clearly defined study period was reported in only two of the six included studies, limiting methodological transparency and comparability. In addition, we emphasized the lack of a uniformly standardized MEBDT protocol in contrast to FEES, which may contribute to methodological variability across studies.

Furthermore, the Conclusion was revised to clearly state that, although the high specificity and positive predictive value of MEBDT support its reliability in confirming aspiration when positive, the moderate pooled sensitivity indicates that a negative MEBDT result cannot reliably exclude aspiration. Accordingly, FEES remains the reference standard, and further prospective validation and expert consensus are required to define the standardized use and appropriate clinical role of MEBDT within diagnostic pathways.

Changes made:

• The Limitations section was expanded to address the small sample size, observational study designs, heterogeneity of clinical practice, limited reporting of study periods, and the lack of standardized MEBDT protocols (page 23, lines 403-411).

• The Conclusion was revised to clarify the implications of moderate sensitivity and negative predictive value and to explicitly state that a negative MEBDT result does not reliably exclude aspiration, reinforcing the role of FEES as the reference standard (page 25-26, lines 428-437).

---

## [Decision Letter · Decision Letter 1]

27 Feb 2026

PONE-D-25-53361R1The Modified Evan’s Blue Dye Test Has High Positive Predictive Value for Diagnosing Aspiration in Tracheostomized Critically Ill Patients: A Systematic Review and Meta-Analysis.PLOS One

Dear Dr. Lovas,

Thank you for submitting your manuscript to PLOS ONE. After careful consideration, we feel that it has merit but does not fully meet PLOS ONE’s publication criteria as it currently stands. Therefore, we invite you to submit a revised version of the manuscript that addresses the points raised during the review process.

We look forward to receiving your revised manuscript.

Kind regards,

Jeyasakthy Saniasiaya, MD, MMed ORLHNS, FEBORLHNS

Academic Editor

PLOS One

**Journal Requirements:**

Reviewers' comments:

Reviewer's Responses to Questions

**Comments to the Author**

1. If the authors have adequately addressed your comments raised in a previous round of review and you feel that this manuscript is now acceptable for publication, you may indicate that here to bypass the “Comments to the Author” section, enter your conflict of interest statement in the “Confidential to Editor” section, and submit your "Accept" recommendation.

Reviewer #2: All comments have been addressed

Reviewer #3: All comments have been addressed

Reviewer #4: All comments have been addressed

2. Is the manuscript technically sound, and do the data support the conclusions?

Reviewer #2: Yes

Reviewer #3: Yes

Reviewer #4: (No Response)

3. Has the statistical analysis been performed appropriately and rigorously? 

Reviewer #2: Yes

Reviewer #3: Yes

Reviewer #4: N/A

4. Have the authors made all data underlying the findings in their manuscript fully available?

Reviewer #2: Yes

Reviewer #3: Yes

Reviewer #4: Yes

5. Is the manuscript presented in an intelligible fashion and written in standard English?

Reviewer #2: Yes

Reviewer #3: Yes

Reviewer #4: Yes

6. Review Comments to the Author

Reviewer #2: Thanks for taking time to review and address the concerns highlighted. Approved for acceptance for publication

Reviewer #3: Thanks for sharing your work with us, and for performing all the suggested comments and revisions for this manuscpit..

Reviewer #4: (No Response)

7. PLOS authors have the option to publish the peer review history of their article (what does this mean?). If published, this will include your full peer review and any attached files.

Reviewer #2: **Yes:** Stephen O Adebola

Reviewer #3: No

Reviewer #4: No

---

## [Author Response · Author response to Decision Letter 2]

9 Apr 2026

Response to Reviewers

We sincerely thank the reviewers for their careful reading of our manuscript and for providing valuable and constructive suggestions, which have greatly helped us improve the clarity and accuracy of our work. All major revisions are highlighted in yellow in the revised manuscript (“Revised Manuscript With Changes Highlighted”), and deleted text has been retained using strikethrough where appropriate.

Clinical claims aligned with meta-analytic results

Reviewer comment: The manuscript should avoid describing MEBDT as a screening test due to moderate sensitivity and low NPV.

Response:

We thank the reviewer for this important comment. We agree and have revised the manuscript to ensure that the clinical interpretation is fully aligned with the meta-analytic findings.

Specifically, MEBDT is now consistently described as an adjunctive rule-in bedside tool rather than a screening test. We explicitly state that negative results cannot reliably exclude aspiration and require further instrumental assessment (e.g., FEES) in high-risk patients.

These revisions have been implemented in the Abstract (final sentence), Introduction (clinical context paragraph), Discussion (first paragraph), Key Points, and Conclusion, and the term “screening test” has been removed or avoided throughout the manuscript where clinically inappropriate.

Positive predictive value (PPV) and prevalence

Reviewer comment: PPV depends on prevalence, and the observed high PPV may be influenced by high prevalence in included cohorts.

Response:

We thank the reviewer for highlighting this important methodological consideration. We agree and have clarified the dependence of predictive values on prevalence in both the Results and Discussion sections.

We now explicitly report the high prevalence of aspiration (approximately 58–82%) in the included studies in the Results (Negative and positive predictive values section, first paragraph).

We further clarify that the high pooled PPV (95%) is influenced by this prevalence in the Discussion (Effectiveness and Limitations of MEBDT section).

In addition, we have included a new figure (Figure 5) illustrating how PPV and NPV vary across a wide range of prevalence values (described in the Results, final paragraph of the predictive values section). We also emphasize that these predictive values are directly applicable only to populations with similar prevalence.

Clinical decision algorithm

Reviewer comment: Proposed algorithms should either be removed or clearly presented as hypothetical.

Response:

We thank the reviewer for this suggestion. In accordance with this recommendation, the clinical decision algorithm has been removed from the manuscript.

Emphasis on heterogeneity and certainty

Reviewer comment: Highlight protocol heterogeneity and low certainty.

Response:

We agree with the reviewer and have strengthened the manuscript accordingly.

We now more explicitly describe the substantial heterogeneity in MEBDT protocols - including differences in bolus consistency, quantity, timing relative to FEES, and patient ventilation status - in the Discussion (Methodology and Implication variability in MEBDT section).

Statistical heterogeneity is reported in the Results (Publication bias and heterogeneity section).

Furthermore, we emphasize that the overall GRADE certainty of evidence is low in the Results (Risk of bias and GRADE assessment section), and we reinforce these limitations in the Strengths and limitations section.

Summary of clinical implications

Response:

We have revised the Key Points and Conclusion to provide a clear and clinically consistent summary.

In the Key Points, we state that:

- MEBDT can serve as a rule-in bedside signal for aspiration when positive

- Negative results cannot exclude aspiration (low NPV)

- Further instrumental assessment is required after negative results

These points are further strengthened in the Discussion (Effectiveness and Limitations section).

In the Conclusion, we summarize that MEBDT should be considered an adjunctive bedside tool, that predictive values may be influenced by prevalence, and that negative results require confirmation with instrumental assessment.

We have also revised the title of the manuscript to better reflect the focus on diagnostic accuracy and the cautious clinical interpretation of findings.

We sincerely thank the reviewers again for their thoughtful and constructive comments, which have helped us improve the clarity and accuracy of our manuscript.

Best regards,

Eszter Szőke

---

## [Decision Letter · Decision Letter 2]

26 Apr 2026

Evaluation of the diagnostic value of the Modified Evan’s Blue Dye Test for assessing aspiration in tracheostomized critically ill patients: a systematic review and meta-analysis

PONE-D-25-53361R2

Dear Dr. Lovas,

We’re pleased to inform you that your manuscript has been judged scientifically suitable for publication and will be formally accepted for publication once it meets all outstanding technical requirements.

Kind regards,

Jeyasakthy Saniasiaya, MD, MMed ORLHNS, FEBORLHNS

Academic Editor

PLOS One

Additional Editor Comments (optional):

Reviewers' comments:

Reviewer's Responses to Questions

**Comments to the Author**

1. If the authors have adequately addressed your comments raised in a previous round of review and you feel that this manuscript is now acceptable for publication, you may indicate that here to bypass the “Comments to the Author” section, enter your conflict of interest statement in the “Confidential to Editor” section, and submit your "Accept" recommendation.

Reviewer #3: All comments have been addressed

Reviewer #4: All comments have been addressed

2. Is the manuscript technically sound, and do the data support the conclusions?

Reviewer #3: Yes

Reviewer #4: Yes

3. Has the statistical analysis been performed appropriately and rigorously? 

Reviewer #3: Yes

Reviewer #4: Yes

4. Have the authors made all data underlying the findings in their manuscript fully available?

Reviewer #3: Yes

Reviewer #4: Yes

5. Is the manuscript presented in an intelligible fashion and written in standard English?

Reviewer #3: Yes

Reviewer #4: Yes

6. Review Comments to the Author

Reviewer #3: (No Response)

Reviewer #4: (No Response)

7. PLOS authors have the option to publish the peer review history of their article (what does this mean?). If published, this will include your full peer review and any attached files.

Reviewer #3: **Yes:** Tahrir Aldelaimi

Reviewer #4: No

---

## [Editor Report · Acceptance letter]

PONE-D-25-53361R2

PLOS One

Dear Dr. Lovas,

I'm pleased to inform you that your manuscript has been deemed suitable for publication in PLOS One. Congratulations! Your manuscript is now being handed over to our production team.

Kind regards,

on behalf of

Dr. Jeyasakthy Saniasiaya

Academic Editor

PLOS One